# Mathematical modeling to reveal breakthrough mechanisms in the HIV Antibody Mediated Prevention (AMP) trials

Daniel B. Reeves[1]*, Yunda Huang[1,2,3], Elizabeth R. Duke[1,4], Bryan T. Mayer[1], E. Fabian Cardozo-Ojeda[1], Florencia A. Boshier[1], David A. Swan[1], Morgane Rolland[5], Merlin L. Robb[5], John R. Mascola[6], Myron S. Cohen[7], Lawrence Corey[1,4], Peter B. Gilbert[1,2,8], Joshua T. Schiffer[1,4,9]

1 Vaccine and Infectious Diseases Division, Fred Hutchinson Cancer Research Center, Seattle, Washington, United States of America, 2 Public Health Sciences Division, Fred Hutchinson Cancer Research Center, Seattle, Washington, United States of America, 3 Department of Global Health, University of Washington, Seattle, Washington, United States of America, 4 Department of Medicine, University of Washington, Seattle, Washington, United States of America, 5 U.S. Military HIV Research Program, Walter Reed Army Institute of Research, Silver Spring, MD USA and Henry M. Jackson Foundation for the Advancement of Military Medicine, Inc., Bethesda, Maryland, United States of America, 6 Vaccine Research Center, National Institute of Allergy and Infectious Diseases, National Institutes of Health, Bethesda, Maryland, United States of America, 7 Division of Infectious Diseases, University of North Carolina at Chapel Hill, Chapel Hill, North Carolina, United States of America, 8 Department of Biostatistics, University of Washington, Seattle, Washington, United States of America, 9 Clinical Research Division, Fred Hutchinson Cancer Research Center, Seattle, Washington, United States of America

* dreeves@fredhutch.org

**Data Availability Statement:** All new data generated are freely available within the manuscript and its Supporting Information files. All

## Abstract

The ongoing Antibody Mediated Prevention (AMP) trials will uncover whether passive infusion of the broadly neutralizing antibody (bNAb) VRC01 can protect against HIV acquisition. Previous statistical simulations indicate these trials may be partially protective. In that case, it will be crucial to identify the mechanism of breakthrough infections. To that end, we developed a mathematical modeling framework to simulate the AMP trials and infer the breakthrough mechanisms using measurable trial outcomes. This framework combines viral dynamics with antibody pharmacokinetics and pharmacodynamics, and will be generally applicable to forthcoming bNAb prevention trials. We fit our model to human viral load data (RV217). Then, we incorporated VRC01 neutralization using serum pharmacokinetics (HVTN 104) and *in vitro* pharmacodynamics (LANL CATNAP database). We systematically explored trial outcomes by reducing *in vivo* potency and varying the distribution of sensitivity to VRC01 in circulating strains. We found trial outcomes could be used in a clinical trial regression model (CTRM) to reveal whether partially protective trials were caused by large fractions of VRC01-resistant (IC50>50 $\mu$g/mL) circulating strains or rather a global reduction in VRC01 potency against all strains. The former mechanism suggests the need to enhance neutralizing antibody breadth; the latter suggests the need to enhance VRC01 delivery and/or *in vivo* binding. We will apply the clinical trial regression model to data from the completed trials to help optimize future approaches for passive delivery of anti-HIV neutralizing antibodies.

computational code used to perform simulations and generate figures is also available (https://github.com/dbrvs/AMP-mechanisms). The data underlying the model-fitting results presented in this study are available from the Military HIV Research Program at Walter Reed Army Research Institute (https://www.hivresearch.org/). Access requires protocol amendments based on patient confidentiality concerns and original study design.

**Funding:** DBR is supported by a Washington Research Foundation (WRF, http://wrfseattle.org/details-eligibility.php) Postdoctoral Fellowship and a Center for AIDS Research (CFAR, http://depts.washington.edu/cfar/find-funding/new-investigator-awards) New Investigator Award (P30 AI027757). Funding was also provided by the National Institute of Health NIAID (UM1 AI068635 to PBG and JTS) The funders had no role in study design, data collection and analysis, decision to publish, or preparation of the manuscript.

**Competing interests:** The authors have declared that no competing interests exist.

## Author summary

Infusions of broadly neutralizing antibodies are currently being tested as a novel HIV prevention modality. To help interpret the results of these antibody mediated prevention (AMP) studies we developed a mathematical modeling framework. The approach combines antibody potency and drug levels with models of HIV viral dynamics, which will be generally applicable to future studies. Through simulating these clinical trials, we found trial outcomes can be used in combination to infer whether breakthrough infections are caused by large fractions of antibody-resistant circulating strains or some reduction in potency against all strains. This distinction helps to focus future trials on enhancing neutralizing antibody breadth or antibody delivery and/or *in vivo* binding.

## Introduction

Interventions that prevent HIV acquisition embody a realistic route to reducing HIV-associated morbidity, mortality, and stigma around the world [1]. Current oral pre-exposure prophylaxis (PrEP) regimens use antiretroviral therapy (ART) to mediate excellent but short-lasting protection against HIV infection [2–4]. A longer lasting PrEP through infusion of broadly neutralizing antibodies (bNAbs) represents a compelling candidate modality [5]. bNAbs display impressive attributes for controlling HIV and may eventually enable host-generated protection if they can be elicited following vaccination [6].

Promising results using bNAbs to control HIV are manifold. Infused bNAbs reduce viral loads in chronically infected individuals [7–9] and extend the time to viral rebound after stopping ART [10, 11]. In nonhuman primate studies, passive infusion of bNAbs before viral challenge repeatedly prevents infection [12–14] and bNAbs administered within 48 hours of infection appear to clear or control virus [15, 16].

Based on these findings, the phase 2b Antibody Mediated Prevention (AMP) trials were designed to assess the prevention efficacy of passively infused bNAbs [17]. Their primary objective is evaluation of the prevention efficacy (PE) of VRC01 [18] (vs. placebo). Previous statistical modeling has estimated wide ranges of PE. Though assumptions in these models cannot be tested currently, PE estimates are inclusive of partial (or incomplete) protection, meaning some treated individuals may be infected [19, 20].

If protection is indeed partial, a crucial objective will be to identify the mechanism of breakthrough infections. To address this, we built on the quantitative mathematical modeling framework of simulated clinical trials [21]. We fit a mechanistic model of natural HIV viral dynamics [22] to data from primary HIV-1 infection in Thailand and East Africa [23] and extended this model to include HIV neutralization by VRC01. To parameterize the initial VRC01 model, we used serum pharmacokinetic (PK) [24] and available *in vitro* pharmacodynamic (PD) characteristics of VRC01 [25]. This approach is generally applicable to any future HIV antibody mediated prevention study. However, we also adjusted our parameterizations because VRC01 concentrations in anatomic sites of HIV exposure may be lower than serum levels, VRC01 binding could be decreased *in vivo*, and available sequences tested against VRC01 may not be representative of circulating strains [26–28]. We explored models analytically in hypothetical 8 week dosing intervals and stochastically in simulated trials.

Trial simulations demonstrated that a combination of clinical outcomes including characteristics of breakthrough viruses (mean and standard deviation of breakthrough virus IC50) and the overall PE, can be used to discern whether most infections are due to decreased potency and/or breadth in the AMP trials. If most breakthrough strains are extremely resistant,

then increasing bNAb breadth or using combination bNAbs could mitigate that challenge [20, 29, 30]. However, if infecting virus is mostly sensitive to VRC01, enhancing potency through better bio-distribution will be a priority. In this way, our mechanistic inference from measurable trial outcomes may help prioritize future development strategies for neutralizing antibodies.

## Results

### Modeling the Antibody Mediated Prevention trials

The antibody mediated prevention (AMP) trials are simultaneous global phase 2b randomized trials HVTN 703/HPTN 081 and HVTN 704/HPTN 085 (ClinicalTrials.gov #NCT02568215 and ClinicalTrials.gov #NCT02716675, respectively). To simulate the AMP trials, we adopted a slightly simplified study design as shown in Fig 1A. Here, each trial contains 3 equally populated arms: placebo, 10 mg/kg, and 30 mg/kg VRC01. Infusion of VRC01 occurs every 8 weeks for 10 total doses and HIV testing occurs every 4 weeks. Our modeling approach involved developing viral dynamics models, incorporating VRC01 PK/PD, simulating trials, and development of a clinical trial regression model (CTRM) capable of distinguishing reduced potency and insufficient breadth as causes of breakthrough infection (Fig 1B).

### A deterministic mathematical model of primary HIV infection recapitulates natural primary infection viral loads and semi-quantitative host-cell dynamics

We designed and validated a deterministic mathematical model of natural HIV primary infection (Fig 2) [22]. Model equations are shown in Eq 5. In this model, susceptible cells $S$ are naturally maintained at homeostasis with birth rate $\alpha_S$ and death rate $\delta_S$. They become infected with rate $\beta$ upon exposure to free virus $V_f$. Infected cells $I_{s,p}$ have a state $s \in [A, L]$ representing active or latent, and a phenotype $p \in [U, P]$ representing unproductively or productively infectious provirus. The death rate of infected cells $\delta_{I_{s,p}}$ depends on their state, with viral cytopathic

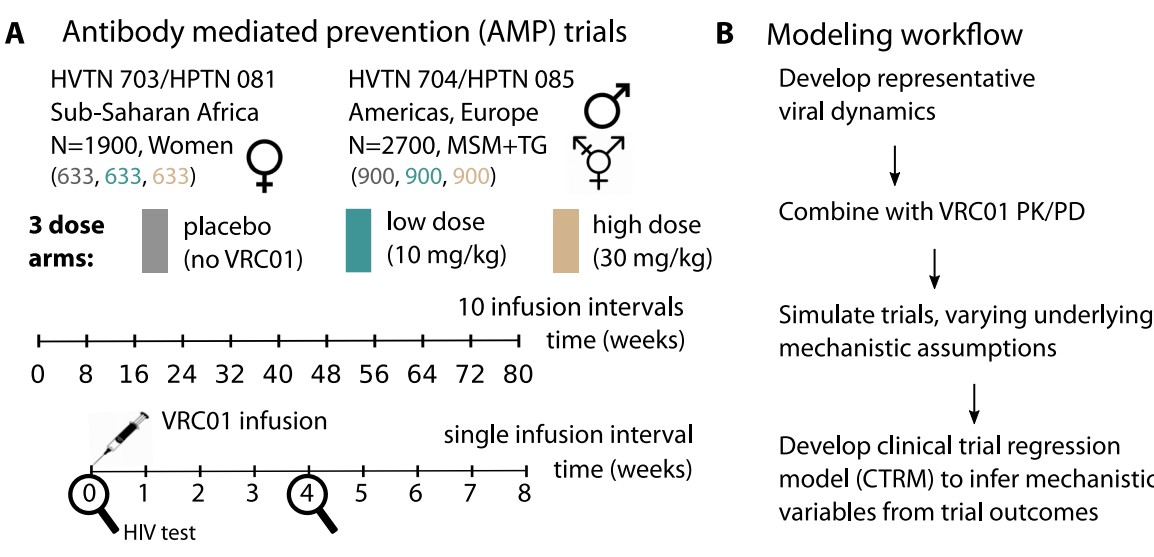

**A** Antibody mediated prevention (AMP) trials

HVTN 703/HPTN 081
Sub-Saharan Africa
N=1900, Women
(633, 633, 633)

HVTN 704/HPTN 085
Americas, Europe
N=2700, MSM+TG
(900, 900, 900)

**3 dose arms:**  placebo (no VRC01)   low dose (10 mg/kg)   high dose (30 mg/kg)

10 infusion intervals
time (weeks)

0  8  16  24  32  40  48  56  64  72  80

VRC01 infusion

single infusion interval
time (weeks)

0  1  2  3  4  5  6  7  8

HIV test

**B** Modeling workflow

Develop representative viral dynamics

↓

Combine with VRC01 PK/PD

↓

Simulate trials, varying underlying mechanistic assumptions

↓

Develop clinical trial regression model (CTRM) to infer mechanistic variables from trial outcomes

**Fig 1. The model implementation of the Antibody Mediated Prevention (AMP) trials.** A) Two parallel trials in African women and North American MSM and TG individuals. Each trial contains 3 equally populated arms, placebo (gray), 10 mg/kg infusions (teal), and 30 mg/kg infusions (tan). Infusions occur every 8 weeks, for a total of 80 weeks and HIV testing occurs every 4 weeks. We refer to an 8 week interval as a 'dosing interval'. B) The model workflow outlines the results.

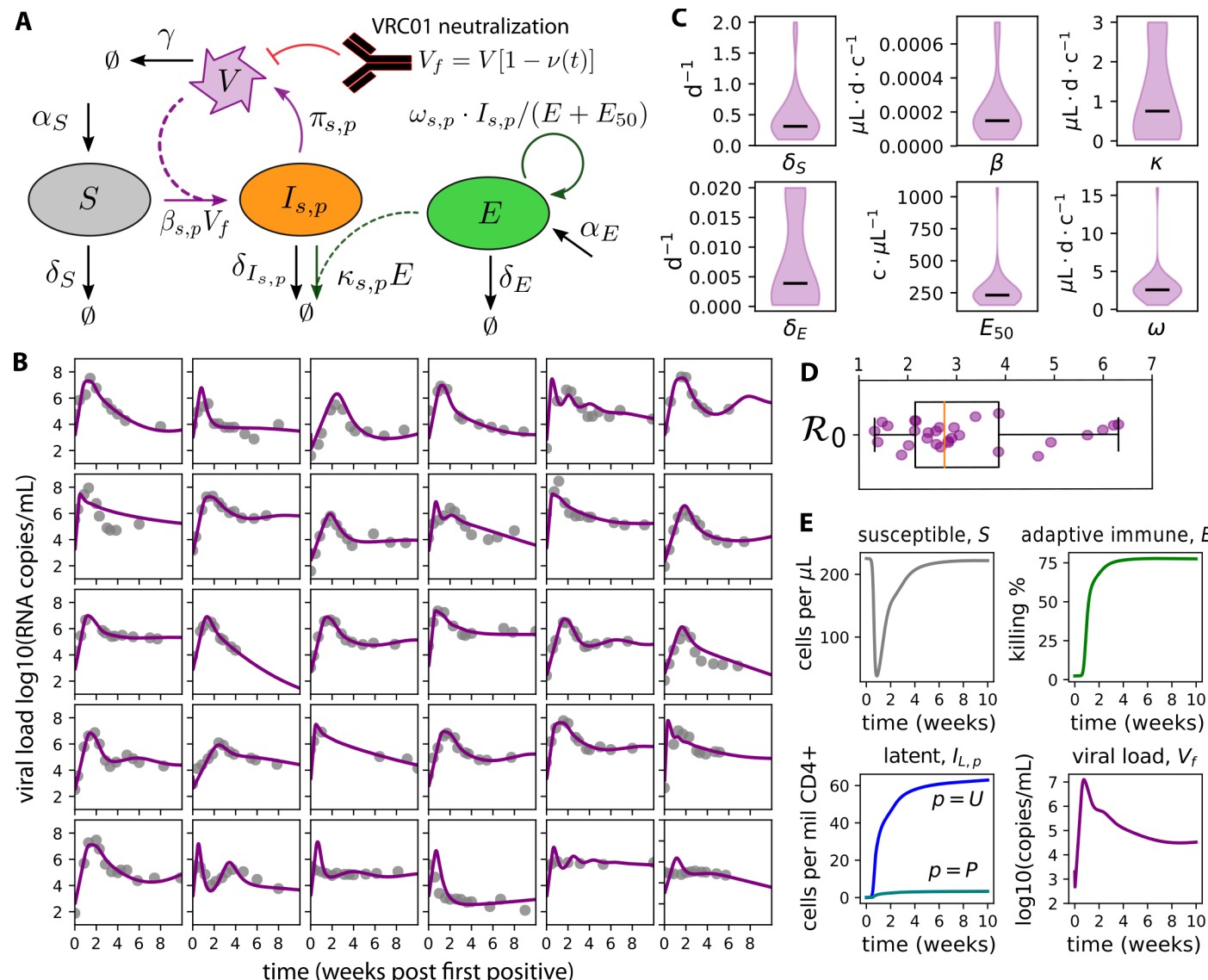

**Fig 2. A mechanistic mathematical model of HIV primary infection including VRC01.** A) Model schematic: susceptible cells $S$ become infected $I_{s,p}$. If the cell is activated $s = A$ and the provirus is productive $p = P$, that cell produces virus with rate $\pi$. An adaptive immune compartment $E$ is recruited to remove infected cells with rate $\omega$, but saturates when $E > E_{50}$. The presence of VRC01 reduces free virus $V_f$. B) Best-fit models of natural HIV dynamics (without VRC01) using data from the RV 217 trial [23] demonstrates satisfactory fit to heterogeneous viral-load kinetics. C) Density plots of distributions of 6 estimated parameters (lines-medians, c-cells, d-days). 8 parameter values were set constant based on past studies; see Table 2 for constant values and initial guesses for estimated parameters. D) Calculated values of the basic reproductive number $\mathcal{R}_0$, the average number of new infected cells generated due to a single infected cell in a fully susceptible population. E) A representative model simulation using mean values from density plots in C recapitulates features of host-cell dynamics during HIV primary infection. Susceptible cells decrease by approximately 200 cells/$\mu$L and reconstitute by viral set-point. Latently infected cells are generated within the first week of infection and contain plausible levels of total HIV DNA (unproductively infectious latent provirus $I_{s = L, p = U}$) and replication competent provirus (productively infectious latent provirus $I_{s=L,p=P}$). Adaptive immune cells $E$ become significant around the time of peak viral load (approximately 10 days after first positive) and adjust viral load set-point.

effects killing active cells rapidly and latent cells having a long (44 month) half-life. Only active, productively infectious cells produce virus (with rate $\pi$). An adaptive immune compartment $E$ is recruited at rate $\omega$ depending on the total number of infected cells, but this recruitment is limited by a saturation constant $E_{50}$. Adaptive immune cells kill actively infected cells with rate $\kappa$ and immunity wanes with rate $\delta_E$. Virus is naturally cleared with rate $\gamma$. When VRC01 is present, free virus is reduced based on the concentration and neutralization potency of VRC01

**Table 1. Sensitivity analysis variables to assess deviations from the initial model predictions.**

| Variable | Range | Name: definition |
|---|---|---|
| $\phi$ | $[1, 10^3]$ | potency reduction factor: fold-reduction in concentration and/or fold-increase in *in vivo* IC50 |
| $f$ | $[0, 1]$ | resistant fraction: fraction of circulating isolates with IC50>50 $\mu$g/mL |
| $A_0$ | $[1, 20]$ | founder cells: the initial number of infected cells |
| $x_{50}$ | $[50, 10^4]$ | maximum IC50: the true maximum IC50 for resistant variants (replaces the detection threshold 50 $\mu$g/mL) |

against the virus. We focus on natural infection initially and include the model for VRC01 neutralization in the following sections.

We parameterized the natural viral dynamics model using human primary infection viral loads from the RV 217 study [23]. The model achieved satisfactory fit (matching peaks, times-to-peak, set-points) to heterogeneous viral load trajectories across all study participants (see Fig 2B). The complete model is not identifiable, so certain parameters were fixed at values based on experimental measurements and past modeling. In particular, latent reservoir parameters were estimated from separate data sources. Table 1 contains all fixed values and initial guesses for estimated parameters. Importantly, this approach does not aim to define the absolute values of viral dynamic parameters, but rather to identify plausible sets of parameters that recapitulate the natural viral dynamics in humans so that these dynamical systems can be studied under the perturbative force of passive immunization with VRC01.

Estimated parameters included the death rate of susceptible cells $\delta_S$, the probability of productively infectious provirus $\tau$ given infection, and several adaptive immune parameters: the killing rate of the adaptive immune compartment $\kappa$, the removal rate of the adaptive immune compartment $\delta_E$ (inclusive of death and exhaustion), the half-maximal saturation constant of the adaptive response $E_{50}$, and the recruitment rate of the adaptive response $\omega$. Population distributions for each of the 6 estimated parameters are organized into density plots showing fits are achieved with relatively narrow parameter ranges across participants (Fig 2C).

We combined estimated parameters with fixed parameters to calculate an approximate basic reproductive number $\mathcal{R}_0$ (see Methods). This value provides the average number of infected cells generated by a single infected cell at the start of infection [31]. If this value is below 1, viral production cannot overcome clearance, and infected cells eventually disappear. Our estimates (Fig 2D) are different from previous estimates [32] as a consequence of using a differently structured model.

Beyond the quantitative fitting procedure, the average parameter values generated several semi-quantitative predictions that were consistent with existing data (Fig 2E). Initial susceptible cell concentrations were approximately 1/4 of typical CD4+ T cell counts in humans, agreeing with observations that not all CD4+ T cells are permissive for HIV replication [33, 34]. These cells were depleted by roughly 200 cells/$\mu$L, roughly agreeing with losses in total CD4+ T cell counts [23]. Soon after detection, numbers of latently infected cells agreed with previous measurements [35, 36] both in magnitude (a total of approximately 100 infected cells per million CD4+ T cells) and in relative proportions of replication competent provirus (approximately 1 in 100 latently infected cells). The relative timing and magnitude of the adaptive immune response coarsely matched experimental observations: our modeled adaptive immune compartment rose several orders of magnitude within 2 weeks following first positive viral load; further, by viral set-point the adaptive immunity has saturated and the infected cell death due to adaptive immunity was comparable to the infected cell death due to cytopathic effects [37–39]—as calculated using the ratio of $\kappa E/(\kappa E+ \delta_I)$.

## The initial model predicts excellent VRC01 protection against HIV acquisition throughout the AMP dosing interval

To predict VRC01 impact on viral dynamics in the dosing arms of the AMP studies, we incorporated a common PK/PD approach into our viral dynamics model.

As documented by two phase 1 trials studying the pharmacokinetic (PK) properties of VRC01 [24, 40], the clearance of VRC01 from serum is bi-phasic with an initial rapid distribution phase and a prolonged secondary elimination phase having a terminal half life of roughly 2 weeks. For simple integration with the viral dynamics model, we modeled the VRC01 PK as

$$\mathcal{Y}(t) = \mathcal{Y}_1 e^{-k_1 t} + \mathcal{Y}_2 e^{-k_2 t} \tag{1}$$

where $\mathcal{Y}_1 + \mathcal{Y}_2$ is the infused VRC01 concentration ($\mu$g/mL of VRC01), and $k_1$ and $k_2$ are the distribution and elimination rates (per day). Our simplified PK model provided excellent fit to the data and best-fit parameters from all individual fits were consistent (S1 Fig).

We also retrieved *in vitro* pharmacodynamic (PD) data from the 47 studies containing HIV/VRC01 neutralization data hosted in the LANL CATNAP database [25] (S2 Fig). A logistic, or 'Hill', function was used to model the fraction of virus neutralized (bound) over continuous VRC01 concentrations (see Methods for derivation). We define the neutralized (or bound) fraction at any time $t$ after a VRC01 infusion as

$$v(t) = \left[ 1 + \left\{ \frac{\text{IC50}}{\mathcal{Y}(t)} \right\}^h \right]^{-1}, \tag{2}$$

so that when the VRC01 concentration is much above the 50% inhibitory concentration (IC50) all virus is neutralized $\mathcal{Y}(t) \gg \text{IC50}$, $v \to 1$, and when $\mathcal{Y}(t) \ll \text{IC50}$, $v \to 0$ and no virus is neutralized. Note, taking a logarithm of this value ($\log v(t)$) admits the instantaneous inhibitory potential (IIP), previously used to calculate the efficacy of antiretroviral therapy [41]. Using Eq 2 and IC50 and IC80 data from S2A Fig we computed each Hill slope $h$. The values of $h$ centered around $h \sim 1$ (S2A Fig), in agreement with previous results [42, 43]).

We simulated PK curves and PD curves separately (Fig 3A & 3B). The lowest concentrations found in simulated AMP interval PK models were roughly 1 $\mu$g/mL (Fig 3A). In comparison, while there are some individual curves with higher IC50s, the median PD curve suggested most PD parameter sets would have an IC50 above 1 $\mu$g/mL (thick black line in Fig 3B). By combining these models (cartoon representation in Fig 3C) we simulated 1000 examples (Fig 3D) to demonstrate the range of expected neutralization given exposures to different CATNAP strains at different times following a VRC01 infusion (25 random examples illustrated to avoid over-plotting).

Next, we modified the viral dynamics to include a reduction in free virus

$$V_f = V[1 - v(t)] \tag{3}$$

so that as neutralized fraction increases, virus decreases. The approximate analytical expression for the basic reproductive number

$$\mathcal{R}_0(t) = \mathcal{R}_0[1 - v(t)], \tag{4}$$

now can be used to coarsely model an exposure at some time $t$ following an infusion with VRC01 (Fig 3E). By combining parameter values from the natural infection model fitting (Fig 2B & 2C) inserted into Eq 17 with the variety of PK/PD values inserted into Eqs 1 and 2, we calculated 300 possible combinations for $\mathcal{R}_0(t)$. Many parameter sets achieved complete protection ($\mathcal{R}_0(t) < 1, \forall t$), but certain parameter sets permitted infection within the first weeks

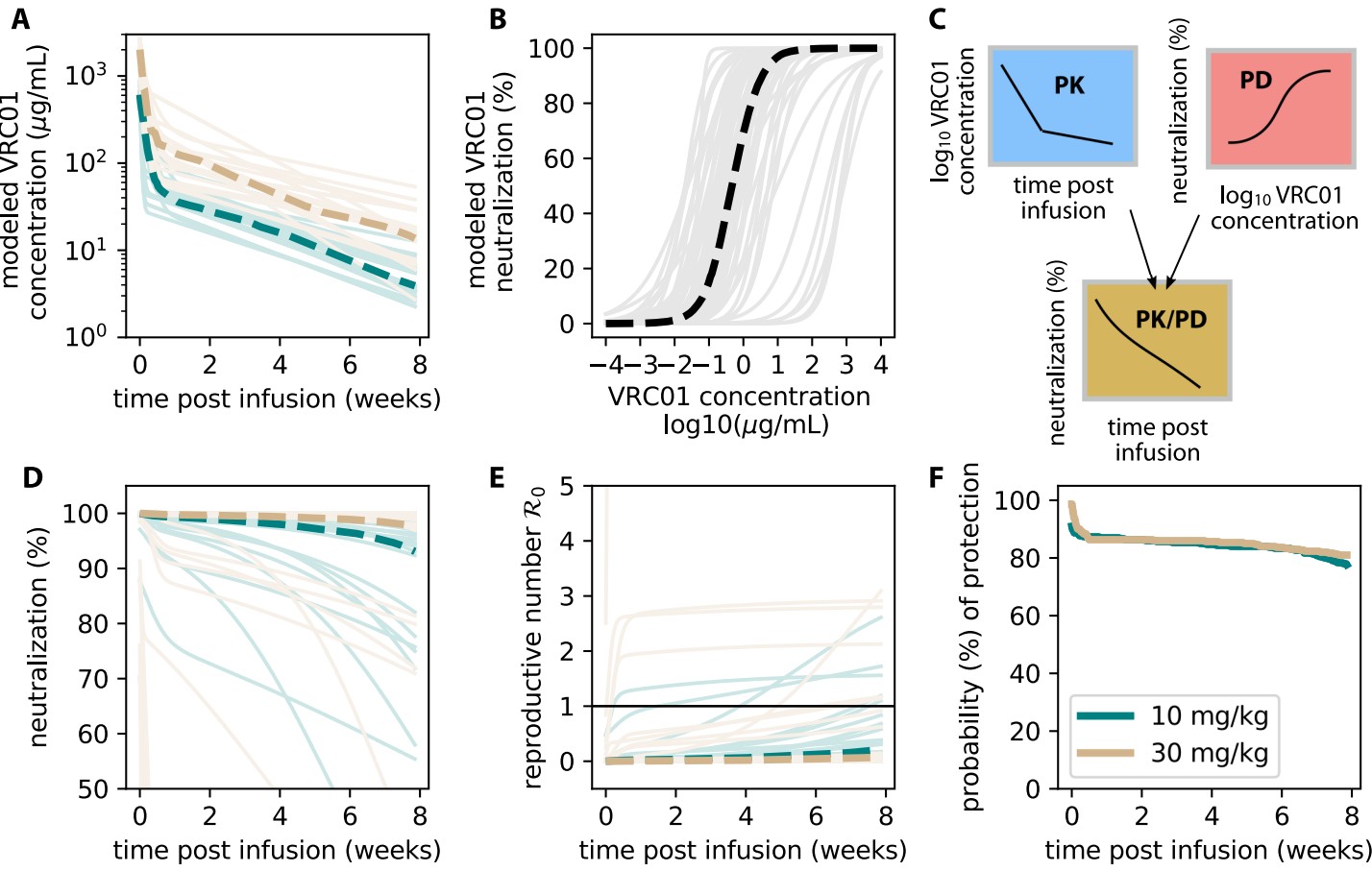

**Fig 3. The serum PK and *in vitro* PD provide an initial estimate for AMP protection.** A) Modeled PK kinetics using estimates from S1C Fig and Eq 1. B) Modeled PD dose-response curves using data from S2A Fig and Eq 2. C) Illustration of the modeling framework: a single PK concentration kinetic curve ($\mathcal{Y}_t$) is combined with a single static PD neutralization (IC50, $h$) to model individual neutralization at some time post infusion (Eq 3). D) 1000 simulations of neutralization during an AMP dosing interval were performed to calculate average quantities (plotted as dashed lines). 25 randomly drawn examples are plotted as lighter lines. E) Modeling the reproductive number $\mathcal{R}_0$ modulated by VRC01 neutralization given an exposure at some time. Each PK/PD curve is rescaled by a single reproductive number (Fig 2, Eq 4). If exposure occurs when $\mathcal{R}_0(t) < 1$ (threshold highlighted by thin black line), protection from infection is likely. Some parameter ranges allow for breakthrough infections, typically late in the dosing interval and in the lower dose arm (10 mg/kg, teal). F) The fraction of simulations having $\mathcal{R}_0(t) > 1$ over time demonstrates that *in vitro* potency measurements and serum concentrations predict excellent prevention efficacy. In all simulations thick dashed lines indicate median values and colors indicate study arm.

after infusion (Fig 3E). If a functional exposure occurs at time $t$ when $\mathcal{R}_0(t) < 1$, that individual is likely to be protected. We use the fraction of simulations where $\mathcal{R}_0(t) > 1$ (thick dashed lines) to estimate protection probability (Fig 3F. These values remained above 75% in both arms even at the end of the dosing interval. We emphasize that these initial model results use serum PK and *in vitro* PD parameters, very likely overestimating protection in the real trials.

## Simulated clinical trials using the initial model predict high prevention efficacy

To study further characteristics of protection including breakthrough viral loads, we adopted a stochastic formulation to study the initial model in a full AMP simulated trial. This model simulates a single infected cell introduced into a discrete number of susceptible cells. The probability of each new cellular infection, viral production, etc. is now considered a stochastic event that occurs with an average rate determined from the deterministic model (See Methods).

Importantly, this allows for extinction (viral clearance) after exposure in the context of bNAbs. As part of model validation, we verified that the first positive after exposure in the placebo arm fell into empirical windows of 7-10 days [44, 45], suggesting the parameterization and stochastic simulator are reasonably representative of early infection. To model bNAb prevention, we required a precise definition for the initial conditions of the simulation. In non-human primate studies, systemic virus is found before clearance with bNAbs, suggesting that bNAbs do not completely prevent infection of all cells [13, 15]. Therefore, we defined 'functional exposures' as those for which virus passes the mucosal barrier and infects at least one susceptible CD4+ T cell. Defined this way, functional exposures have a >90% chance of causing infection in the natural stochastic model. We used the incidence rates and population sizes of the AMP cohorts to estimate the expected number of functional exposures in each study arm. For example, a 3% incidence rate per year (as estimated for for the Americas and Europe [17]) with a 500 individual trial for 80 weeks admits approximately 40 functional exposures (Fig 4A). We assumed incidence was identical in each dosing arm and simulated exposures as uniform over time, drawing the parameters for each exposure from a set of viral dynamic (Fig 2C), PK (S1C Fig), and PD (S2A Fig) parameters.

In a representative simulated trial example with the initial model (Fig 4), VRC01 had a strong effect on total prevention efficacy and a dose-dependent prevention efficacy was observed: $PE = 1 - 6/36$, or 83%, in the low (10 mg/kg) dose arm and $PE = 1 - 3/36$, or 92%, in the high (30 mg/kg) dose arm, respectively (see Eq 20). The approach of viral dynamics also highlights a possible challenge for AMP. In theory, some individuals will receive another dose of VRC01 after infection but before first positive, thereby suppressing viral load and presenting a complicated scenario for confirmatory measurements. In some simulations we observed undetectable viral loads and latently infected cells, representing the possibility for occult infection.

## A framework to detect the mechanistic causes of deviations from the initial model predictions

There are several reasons why true prevention efficacy in the AMP studies might be lower than the estimated PE from initial model simulations. Therefore, we studied four mechanistic hypotheses for reduced prevention efficacy to determine whether deviations from the initial model could be inferred from characteristics of breakthrough infections (summary in Table 1).

First, concentration of VRC01 at sites of HIV exposure may be lower than concentrations measured in serum. Additionally, *in vivo* neutralization in mucosa might theoretically be reduced due to anti-idiotype antibodies or competitive exclusion [46]. Both mechanisms represent a global reduction in VRC01 neutralizing impact based on deviations from serum PK and/or *in vitro* IC50 [27]. To study these mechanisms, we introduce the 'potency reduction factor' $\phi$. Mathematically, this parameter decreases the ratio of VRC01 concentration to IC50, multiplying the term $\frac{IC50}{\mathcal{Y}(t)} \times \phi$. This simple scaling is plausible because a significant correlation has been identified between *in vitro* IC50 and prevention efficacy in non-human primate studies [27, 28] and two studies demonstrated that VRC01 in human serum retained its neutralization ability [40, 47]. Based on evidence from non-human primate studies, in which antibody concentrations in the vaginal fluid were 1-2 orders of magnitude lower than serum concentrations [26], we varied the potency reduction from 1 (no change from initial model) to 1000.

Second, circulating strains in the AMP trials may not be well-described by the laboratory isolates collected in the CATNAP database. In other words, VRC01 may have less breadth against isolates in the AMP study than those in the database. To study this impact, we

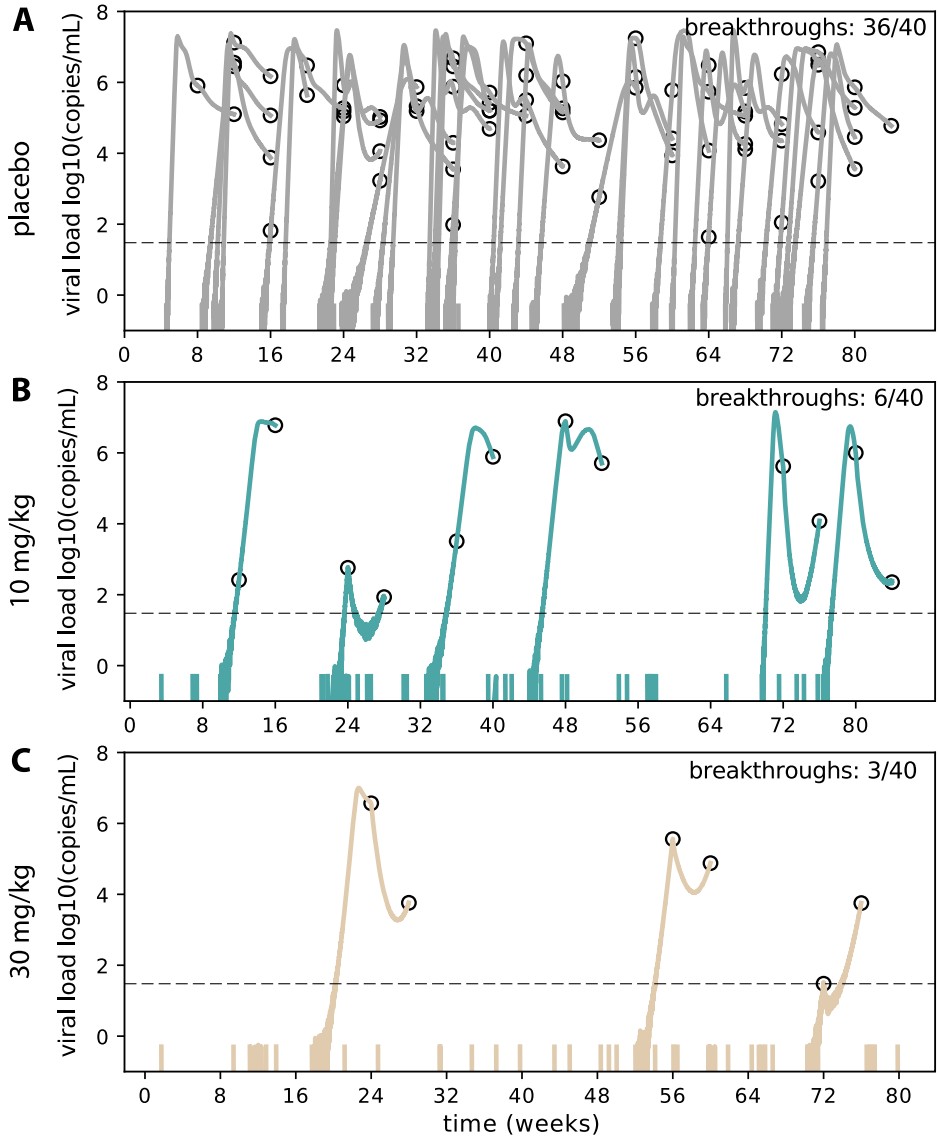

**Fig 4. A representative AMP simulation using the initial model (serum PK and *in vitro* PD).** For each dosing arm in the trial (placebo, 10 mg/kg, and 30 mg/kg, respectively), we simulated 40 functional exposures (based on a 3% incidence rate per year) and plotted viral load data: colored solid lines are simulated viral loads and open circles represent viral load measurements given the sampling frequency of the AMP trials. Dashed black line indicates a limit of detection at 30 copies/mL. Each exposure is initialized with a single infected cell carrying a randomly drawn viral strain (IC50) and a VRC01 concentration which follows randomly drawn participant PK kinetics. To normalize correctly, we account for the fact that in the placebo model (A), some functionally exposed participants naturally clear the infection due to variability in host immune parameters and stochastic simulation. Many functional exposures are blocked by VRC01. The trial admits excellent prevention efficacy: $PE = 1 - 6/36$, or 83%, in B) the low dose arm and $PE = 1 - 3/36$, or 92%, in C) the high dose arm.

parameterized a distribution containing sensitive and resistant strains. While recognizing resistance is a continuum, we chose sensitive and resistant strains as those with IC50 <1 and >50 $\mu$g/mL, respectively. This definition alludes to the definition of breadth as the fraction of viruses with an IC50 above the detection threshold [48]. We then chose a bi-modal distribution to parameterize exposing sequences. A fraction *f* of IC50s were drawn from a lognormal distribution centered on $IC50 = 10^3$ $\mu$g/mL with variance 1 log and the remaining $1 - f$ were

drawn from a lognormal distribution centered on $IC50 = 10^{-2}$ $\mu$g/mL with variance 1 log. We varied the 'resistant fraction' $f$ from 0 (no resistant strains) to 1 (all resistant). The values for distribution modes allow for edge cases where at the low end $f = 0$, most sequences are neutralized easily, and at the high end most sequences can infect over the entire dosing window (given initial infusion concentrations usually less than $10^3$ $\mu$g/mL.

Third, viral inoculum size [49] and/or transmission-site anatomy [50] appears to affect the number of founder viruses. To address the possibility that functional exposures with more than one infected cell could overwhelm antibody protection and reduce stochastic burn-outs, we varied the number of active, productively infected 'founder cells' $I_{A,P}(0)$, shorthanded as $A_0$ from 1-20.

Lastly, available *in vitro* data have a detection threshold and do not quantify IC50 values above 50 $\mu$g/mL. Thus, it is unclear exactly how much VRC01 would be needed to neutralize these strains. In addition, 'incomplete neutralization' has been documented [51, 52]. To model incomplete neutralization, we introduced $x_{50}$ to represent the maximum IC50, which we varied from the detection threshold of 50 $\mu$g/mL up to 10,000 $\mu$g/mL.

Global sensitivity analyses of simulated clinical trials were employed to test the impact of the sensitivity analysis variables on clinical outcomes. 100 parameter combinations were created using Latin hypercube sampling (LHS) from the ranges of the four sensitivity analysis parameters (Table 1). For each dosing arm, 100 functional exposures were simulated and five outcomes were recorded. Variable 1 is prevention efficacy (PE). Variables 2 and 3 are the mean and standard deviation of the first positive viral load in all detected breakthrough infections. The last variables (4 and 5) are the mean and standard deviation IC50s of breakthrough viruses as would be measured by an *in vitro* assay. These outcomes are referred to as measurable because they will be available in a post-hoc fashion from the AMP trials [17].

By correlating the values of the four sensitivity analysis variables against the five measurable trial outcomes Fig 5A & 5B, we observed several important relationships. The variable correlated most strongly to prevention efficacy was the resistant fraction $f$. The standard deviation of breakthrough IC50s was most correlated with the potency reduction factor $\phi$. Both resistant fraction and potency reduction factor impacted the mean IC50 of breakthrough strains comparably. The maximum IC50 $x_{50}$ and the initial number of infected cells $A_0$ did not correlate strongly with any clinical outcomes.

To help interpret the impact of potency reduction factor and resistant fraction, we examined all breakthrough IC50s in a given trial (S3 Fig). Decreased potency allowed a wider distribution of strains (inclusive of lower sensitivity) to infect: thus the variability of breakthrough IC50 increased. Raising the resistant fraction increased the mean IC50 of breakthrough strains and decreased the variability.

## A clinical-trial regression model (CTRM) accurately distinguishes partial efficacy due to reduced global potency vs. insufficient breadth against circulating strains

Based on the finding that outcomes were distinctly correlated with different biological variables (Fig 5), we developed a framework to assess the reverse correlation. That is, we sought to use the measurable trial outcomes to distinguish which unobservable biological variable was most-responsible for partial trial efficacy.

Using the global sensitivity analysis results, we performed linear multivariate multiple regression [53] using the measurable outcomes as the predictor variables and the sensitivity analysis variables as the dependent variables. Using only the five measurable outcomes of each trial, we assessed our prediction of potency reduction and resistant fraction using a cross-

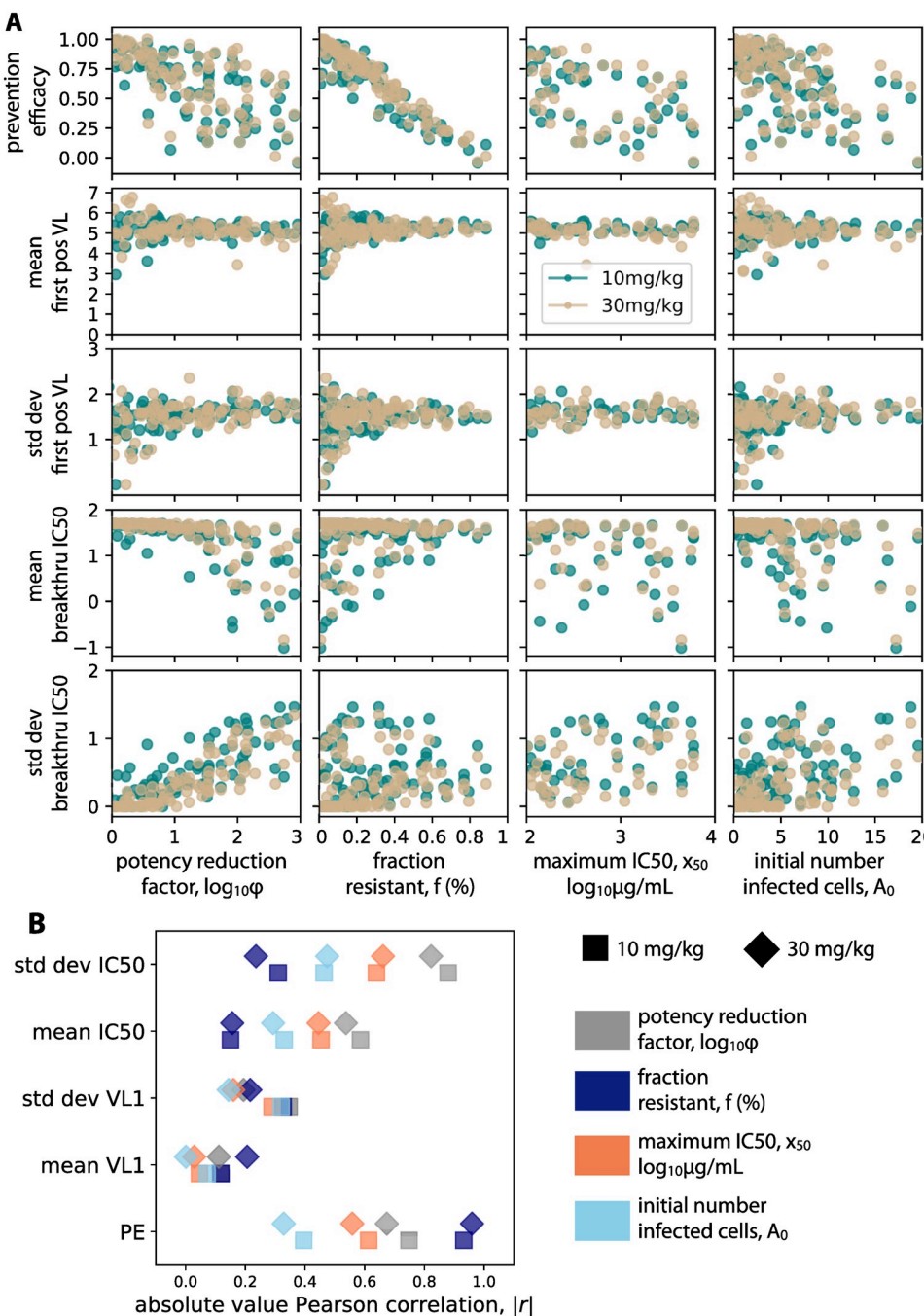

**Fig 5. Global sensitivity analysis links unknown biological variables to measurable trial outcomes.** A) Results from 100 simulations of each dosing arm correlating sensitivity analysis variables (see Table 1 for definitions) against measurable trial outcomes. For each simulation, a value of each sensitivity analysis variable was chosen from a LHS sample. 100 functional exposures were simulated in each trial. Correlations agree in both dosing arms—teal 10 mg/kg, tan 30 mg/kg. B) Absolute values of Pearson correlation coefficients (marker shape indicates dosing arm) show that different variables correlate more strongly with different outcomes. Potency reduction factor and resistant fraction are the most strongly correlated overall, but each correlates more strongly with a different sensitivity analysis variable: potency reduction factor most strongly correlates with the standard deviation of breakthrough virus IC50, whereas resistant fraction correlates most strongly with prevention efficacy.

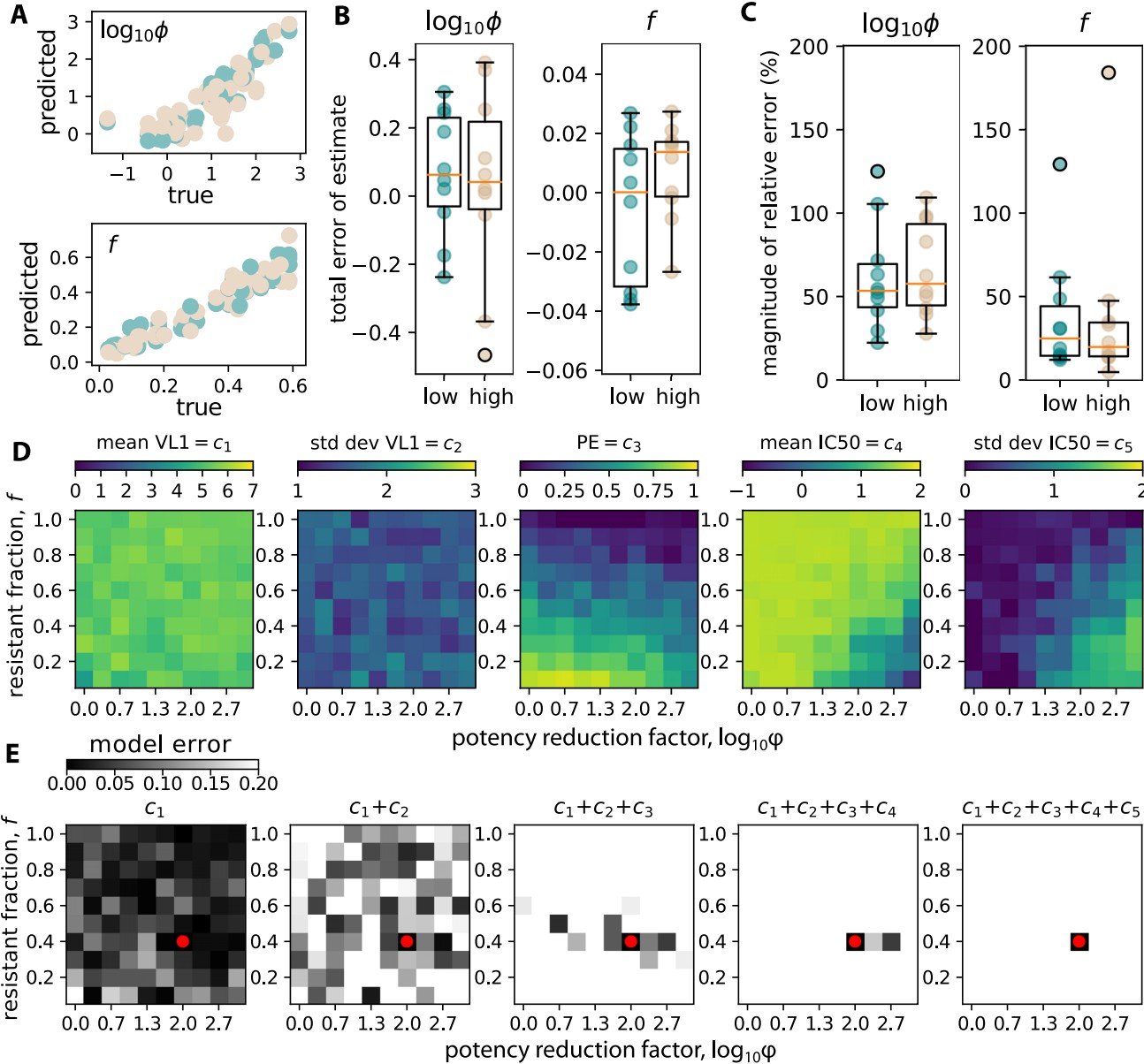

**Fig 6. A clinical trial regression model (CTRM) uses trial outcomes to infer potency reduction and resistant fraction.** We simulated 100 trials in each dose arm, and performed 5-fold cross validation. We used sets of 40 randomly selected trials to train the CTRM, and then tested the CTRM to predict potency reduction and resistant fraction only using measurable trial outcomes from 10 different randomly selected trials. Errors were then averaged across 5 replicates. A) True vs predicted values for both variables. B) Total error (true−predicted) indicates reasonably unbiased estimates. C) Relative error (|true−predicted|/true × 100) indicates estimation error is typically at or below 50%. D) Heat maps indicate no single variable distinguishes reduced potency from increased resistance. E) Illustration of sequential inference using trial outcomes using an example simulation with a certain potency reduction and resistant fraction ($\phi = 100$, $f = 0.4$, red dot). The inference proceeds by applying trial outcomes $c_i$ sequentially to filter out unlikely simulated parameter combinations. Darker boxes indicate higher likelihood of a certain parameter combination. First positive viral dynamic outcomes (mean $c_1$ and std dev $c_2$) do not discriminate strongly. However, by filtering by prevention efficacy $c_3$ and IC50 outcomes (mean $c_4$ and std dev $c_5$), the true value is identified correctly.

validation train/predict scheme. We randomly chose 40 simulations to train, and 10 to predict. This was repeated 5 times and errors were averaged. Training and testing was repeated for each dose arm. Estimates were reliably accurate (see true vs. predicted in Fig 6A). Absolute error (true−predicted, Fig 6B) was mostly unbiased, with median values for potency reduction

slightly below 0. Relative error (|true−predicted|/true × 100) indicates estimation error is typically at or below 50% with some outliers (Fig 6C).

To demonstrate the approach intuitively, we plotted heat-maps relating the potency reduction factor and resistant fraction to each of the five measurable outcomes (Fig 6D). No single outcome is sufficient to identify the value of both sensitivity analysis variables. However, in practice, by using the value of each outcome found from an example trial, we can sequentially constrain the possibilities for the two values, until a reasonable estimate is developed. In Fig 6E, we perform this process by beginning with a simulation with a certain potency reduction and resistant fraction ($\phi = 100$, $f = 0.4$, red dot). This value of potency reduction was chosen because it is plausible based on nonhuman primate bNAb concentration studies [26]. Then, using each heat map from Fig 6D, we calculated the percent error error where more precise estimates are represented by darker shades. Clinical outcomes $c_i$ are applied sequentially. Viral dynamic outcomes (first positive mean $c_1$ and std dev $c_2$) rule out some parameter combinations, but are not very discriminatory. However, prevention efficacy $c_3$ and IC50 outcomes (mean $c_4$ and std dev $c_5$) strongly constrain the possible parameter combinations. Gray shades indicate the likelihood of parameter combinations Fig 6B & 6C. After all 5 outcomes have been used to filter out unlikely parameter combinations, the darkest square correctly identifies the true value.

## Discussion

To aid the interpretation of the AMP trials, we developed a mathematical modeling framework to simulate the clinical trials. Our model is informed by numerous conceptual (e.g., Refs. [54–56]) and mathematical (e.g., Refs. [57–60]) models of viral dynamics during early HIV infection. It satisfies several important criteria relevant to simulating the AMP trials by recapitulating diverse HIV viral load dynamics at the individual and population levels. Further, it qualitatively agrees with measurements of host-cell levels including total infected cells, latently infected cells (containing both replication competent and total HIV DNA levels), and the relative impact and timing of adaptive immunity. We integrated a combined PK/PD model of VRC01 and developed both an analytical approximation and a full stochastic simulation of the prevention efficacy of VRC01 in the AMP studies. Using available serum PK and *in vitro* PD values, we predicted high prevention efficacy comparable to previous oral PrEP studies [3].

However, this initial model may not directly translate to the *in vivo* activity of VRC01 in the AMP studies. Thus, instead of attempting to predict trial outcomes based on unknown biology, we developed a clinical trial regression model (CTRM) framework to infer the unknown biology and interpret the mechanistic causes of a partially protective bNAb. Our approach distinguishes breadth from potency, two concepts that are interrelated. We define potency as a global reduction in neutralization against all challenge viruses—in other words, a scaling of the average IC50, whereas breadth is a characteristic of a bNAb against the circulating strains—in other words, a spread of the IC50s. When breadth increases, the bNAb becomes better at neutralizing some viruses. When potency increases, it becomes better at neutralizing all viruses. Potency reduction could be due to decreased VRC01 concentration at relevant sites and/or decreased *in vivo* neutralization activity of VRC01. Breadth reduction could be due to a population of circulating viruses specifically resistant to VRC01.

We proceed to highlight how measured quantities from the AMP studies can be combined to distinguish between breadth and potency, inferring mechanistic biology and creating a pathway for optimizing sequel trials. Specifically, simulated trials with weak breadth, i.e. the circulating population contains many very resistant viruses (IC50 $>50\mu g/mL$), lead to low preventative efficacy with a high mean and low standard deviation of breakthrough virus IC50.

Simulated trials with weak *in vivo* potency, i.e. breakthrough is possible even though *in vitro* measurements of IC50s are below infused concentrations, lead to a lower mean and high standard deviation of breakthrough virus IC50 (summary in Fig 6D).

There are several important limitations to our approach. To simplify trial logistics, we assume perfect adherence to visit schedule. Further, while we implicitly assume viral infectivity and transmission biology is equivalent in each trial arm; we do not make predictions across trials where cohort makeup and distribution of virus clades are different.

In the stochastic implementation of the model, we assume that VRC01 prevents expansion after infection of a single cell, rather than inducing complete protection against infection of any cells. This is based on detailed nonhuman primate studies where bNAb-mediated protection involved systemic measurable virus in tissue compartments [13, 15]. No additional timescale for mucosal passage was built in to these simulations, but that timescale is typically on the order of hours, which would only minimally shift the present results [61]. In previous modeling of bNAb prevention [43], authors determined a per-virion probability of infection. Our approach mirrors this in assigning an effective probability of infection per infected cell. Yet, our viral dynamics approach is distinct because it defines a non-linear and time-varying relationship between bNAb levels and infection probability.

We only model how VRC01 can protect against 'functional exposures', those where at least 1 cell is infected in the new host. The regime in which all virions in an exposure are completely neutralized is indirectly captured by our model as cases where $\mathcal{R}_0(t) \ll 1$. As defined, such functional exposures typically proceed to viremia naturally. This is important because our viral dynamic parameter estimates are necessarily from individuals who were naturally infected, which may not represent all HIV exposures (i.e. estimates are conditional on infection [59]). As a consequence, our model does not account for the rare (<1%) per-coital transmission probability of HIV in the natural setting ([56, 62]). Further, HIV incidence rates have been estimated in these trial populations, and we could calculate a fairly reliable estimate of the number of functional exposures expected in each arm.

We assume VRC01 reduces free virus based upon multiple studies showing infusion of VRC01 during chronic infection causes viral load to decay at a rate comparable to the decay following administration of ART [10, 63]. It's possible these data are not well-enough resolved, and that VRC01 does increase the infected cell death rate via antibody-dependent cellular cytotoxicity (ADCC) [9]. For instance, the partially effective RV144 Thai trial suggested ADCC could affect protection [64], but other studies have shown minor ADCC impact [5]. If ADCC enhances protection, our model predictions will represent a lower (rather than upper) boundary on prevention efficacy.

Another limitation is that we do not explicitly model the genetic diversity of HIV. Therefore, we cannot address HIV escape during infection. Within-host escape was documented when VRC01 was infused during chronic HIV infection [63]. However, primary infection may be different. Founder virus populations are likely to be much smaller in absolute size and the reduced genetic diversity makes similar rapid escape unlikely. On the other hand, if there are multiple founder viruses, immediate selection for a more resistant variant is theoretically possible. If the AMP trials result in substantial multi-strain infections, our model will be updated to accommodate. For instance, it might be possible to disentangle multiple founder infections based on an updated model that replicates observed variability in set-point due to multiple founders [65].

Our main results on mechanistic inference are applicable for other mechanistic models because specific viral dynamics are not essential to discrimination between mechanisms (see Fig 6E). Most importantly, as long as there is a monotonic relationship between infection

probability and bNAb concentration, PE, mean IC50 and standard deviation of IC50 will behave qualitatively as in Fig 5A), making some inference possible.

In summary, we developed a framework for simulating HIV prevention trials with neutralizing antibodies. This framework allows direct combination of viral dynamics, antibody pharmacokinetics, and antibody pharmacodynamics—including estimates of ranges of parameters for all. Our clinical trial regression model (CTRM) was able to differentiate the two most plausible drivers of breakthrough infection in a simulated partially protective trial: a global reduction from serum concentrations and *in vitro* predicted neutralization vs. insufficient breadth against circulating strains. This approach could help prioritize future development strategies for neutralizing antibodies. If breakthrough viruses possess a wide spectrum of IC50s, then future studies will need to raise *in vivo* potency, by enhancing distribution to infection sites and/or mitigating decreases in *in vivo* binding. However, if breakthrough strains possess similar and high IC50s, future trials will need to enhance breadth to accommodate circulating strains.

# Materials and methods

## Ethics statement

This paper uses de-identified data obtained previously and no new observations requiring patient consent or institutional review board approval have been performed.

## Data and computational code availability

Code and data to replicate all analyses are hosted at https://github.com/dbrvs/AMP.

## Mechanistic mathematical model for primary HIV infection in the context of bNAb intervention

We designed a mathematical model to describe primary HIV infection in the context of protection with broadly neutralizing antibodies (bNAbs). Our model (Fig 2A) has evolved from the canonical models of viral dynamics [66–68] and was developed previously to model SHIV kinetics [22].

The model contains susceptible cells (*S*; likely CCR5+/CD4+ T cells), which are produced constantly with rate $\alpha_S$ and are removed with rate $\delta_S$. Free virus $V_f$ infects these cells and generates infected cells $I_{s,p}$. These cells have a cell state and phenotype (subscript *s* and *p*). We use a vector notation to express the four possible cell states, that is $I_{s,p} = [I_{A,P}, I_{A,U}, I_{L,P}, I_{L,U}]$, where active/latent is denoted by *A*, *L* and productive/unproductive is denoted by *P*, *U*. The infectivity and subsequent designation of cell state and phenotype is given by $\beta_{s,p} = \beta \times [\tau(1 - \lambda), (1 - \tau)(1 - \lambda), \tau\lambda, (1 - \tau)\lambda]$. Only productively infected cells produce virus, so $\pi_{s,p} = [\pi, 0, 0, 0]$. Both productive and unproductive infected cells die rapidly based on viral cytopathic effects and/or bystander killing [69, 70]. Latently infected cells persist with extremely long half-lives [71], but for this model we ignore reactivation from latency because it is rare and thus negligible during primary infection. Therefore, $\delta_{I_{s,p}} = [\delta_I, \delta_I, \theta_L, \theta_L]$.

The adaptive immune system is modeled as an additional effector cell compartment *E*, which is inclusive of CD8+ T cells and other cells [72, 73]. Effector cells help control viral replication by killing all non-latent cells with rate $\kappa_{s,p} = [\kappa, \kappa, 0, 0]$. The recruitment rate of new adaptive immunity ($\omega_{s,p} = [\omega, \omega, 0, 0]$) governs this recruitment and a saturation constant $E_{50}$ limits recruitment. Effector cells are naturally produced with the rate $\alpha_E$ to mimic the extremely low precursor frequency [74] and are also naturally removed (through death or exhaustion) with rate $\delta_E$.

The model is expressed as ordinary differential equations,

$$\dot{S} = \alpha_S - \delta_S S - \beta S V_f$$

$$\dot{I}_{s,p} = \beta_{s,p} S V_f - \delta_{I_{s,p}} I_{s,p} - \kappa_{s,p} I_{s,p} E$$

$$\dot{E} = \alpha_E + \omega_{s,p} \cdot I_{s,p} \frac{E}{E + E_{50}} - \delta_E E$$
(5)

$$\dot{V} = \pi_{s,p} I_{s,p} - \gamma V - \beta S V_f,$$

where time derivatives are denoted by the overdot, $\beta = \Sigma_{s,p} \beta_{s,p}$, and $a \cdot b$ denotes the inner product. We incorporate the effect of VRC01 into the model above by adjusting the fraction of free virus $V_f = V[1 - v(t)]$ where $v(t)$ is the fraction neutralized VRC01 (described below).

## Natural primary infection viral load data

Human primary infection viral load data were collected during the RV217 study; complete details are found in Ref. [23]. The study followed 2276 high-risk volunteers, and participants having detectable HIV-1 were followed such that plasma HIV-1 RNA could be quantitated twice weekly. For modeling purposes, we used data from 30 individuals from Thailand and Uganda whose viral load was observed at least 5 times, who had a single viral load measurement above $10^5$ copies/mL and had viral load measurements for viremia for at least 10 weeks following first positive viral load.

## Estimation of model parameters for natural HIV primary infection

We fit the model to the RV217 viral load data using SciPy's `optimize` package, which uses a least-squares Levenberg-Marquardt approach equivalent to maximum likelihood with a normally distributed variance for the data. 6 out of 14 parameters are fit, with the remaining values fixed (see Table 2 for fixed values and initial guess of fit parameters).

A more complete discussion of the identification of parameter values is provided in a previous work using a similar model [22]. Briefly, viral clearance rate $\gamma$ was estimated from a

**Table 2. Summary of model parameters found in the model, Eq 5.** Parameters in () indicate initial values for estimation; others were held fixed.

| Parameter | Value | Meaning | Dimensions | Reference |
|---|---|---|---|---|
| Constant | | | | |
| $\alpha_S$ | 70 | Susceptible cell production rate | cells $\mu$L$^{-1}$day$^{-1}$ | [75] |
| $\tau$ | 0.05 | Productively infectious probability | unitless | [69] |
| $\delta_I$ | 0.8 | Infected cell death rate | day$^{-1}$ | [76] |
| $\pi$ | $5 \times 10^4$ | Viral production rate | virions cell$^{-1}$day$^{-1}$ | [77] |
| $\gamma$ | 23 | Viral clearance rate | day$^{-1}$ | [78] |
| $\alpha_E$ | $10^{-5}$ | Adaptive immunity production rate | cells $\mu$L$^{-1}$day$^{-1}$ | [74] |
| $\lambda$ | $10^{-4}$ | Latently infected probability | unitless | [79] |
| $\theta_L$ | $5.2 \times 10^{-4}$ | Clearance rate of latently infected cells | day$^{-1}$ | [80] |
| Fit | | | | |
| $\beta$ | $(10^{-4})$ | Total viral infectivity | $\mu$L virions$^{-1}$day$^{-1}$ | [75] |
| $\delta_S$ | (0.2) | Susceptible cell removal rate | day$^{-1}$ | [75] |
| $\delta_E$ | (0.002) | Adaptive immunity removal rate | day$^{-1}$ | [81] |
| $\kappa$ | (0.3) | Adaptive immunity killing rate | $\mu$L day cells$^{-1}$ | [82] |
| $\omega$ | (1.6) | Adaptive immunity recruitment rate | $\mu$L day cells$^{-1}$ | [81] |
| $E_{50}$ | (250) | Adaptive immunity limiting concentration | cells $\mu$L$^{-1}$ | [83] |

human apheresis experiment. Infected cell death rate $\delta_I$ was estimated from human viral load decay data after initiation of ART. Viral production was estimated from a single cycle SIV experiment in macaques. Productive virus fraction was estimated in *ex vivo* cultures of human tonsil tissue. Susceptible cell rates ($\alpha_S$ and $\delta_S$) were estimated from viral load data in a human treatment interruption trial. Importantly, that model has the same structure as our own. Adaptive immune rates are difficult to estimate because the model does not specify a specific cell phenotype. Human experiments have estimated anti-HIV T cell precursor frequencies, which we use to estimate $\alpha_E$. *in vivo* microscopy was used to estimate CD8+ T cell killing rates, which we use to estimate $\kappa$. Maximal fractions of virus-specific CD8+ T cells in mouse LCMV experiments are 20–70%, which we combine with typical human CD8+ T cell concentrations to estimate $E_{50}$. The recruitment and death rates of the adaptive response are estimated from a T cell proliferation study in HIV-infected humans.

The rates governing susceptible and effector cells are the least well-understood, as well as the source (in our model) of the large variation in viral set-point equilibrium. Thus when fit, these values vary the most across participants. For instance, see the ranges of values of $\kappa$ and $\delta_E$ shown in Fig 4C. There are some correlations among parameters and for the purposes of this work we do not claim to have identified absolute values. Rather, we have developed a variety of parameter sets for viral dynamics to be adjusted in the presence of VRC01.

For the deterministic model fitting, the initial viral load $V(t = 0)$ is specified at a 30 copies/mL detection limit. Susceptible cells and adaptive immunity are initialized to equilibrium values, $S(0) = \alpha_S/\delta_S$, which assumes no real T cell depletion before first positive, and $E(0) = \alpha_E/\delta_E$, which assumes no HIV-specific immune response before first positive. The infected cells initial conditions are calculated using a quasi-static equilibrium so that the total number of infected cells is $I(0) = V(0)\gamma/\pi$. Then, we use the proportions to calculate $I_{s,p} = I(0)\beta_{s,p}/\beta$.

## Modeling VRC01 pharmacokinetics (PK)

A detailed pharmacokinetic (PK) model for VRC01 was published by Huang et al. [24]. Here we adapt a simpler analytical formulation so that the concentration of VRC01 at any time $t$ after a dose can be expressed as

$$\mathcal{Y}(t) = \mathcal{Y}_1 e^{-k_1 t} + \mathcal{Y}_2 e^{-k_2 t}. \tag{6}$$

The terminal half-life of VRC01 can be calculated using the second phase decay constant $t_{1/2} = \ln(2)/k_2$.

Using SciPy's `optimize` package, we estimated all four parameters based on data from HVTN 104 in which HIV-uninfected adults received multiple-dose intravenous VRC01 at 10 and 30 mg/kg every 4 or 8 weeks. Each individual (n = 12) received 3 infusions (S1A Fig). We used data for each infusion to estimate 32 and 31 parameter sets for each dose, respectively. S1B Fig shows all fits, and S1C Fig the estimated parameters. These parameter sets could be used to simulate continuous VRC01 trajectories (as in Fig 3A).

## Curation of pharmacodynamic data

We use tabulated pharmacodynamic (PD) data including measurements of IC50 and IC80 from 782 viral isolates as tabulated in the LANL CATNAP database [25]. Some of these isolates are repeated, but measurements are not identical, so we included all data. Using IC50 and

IC80 we calculated Hill coefficients for each isolate as

$$h = \frac{-\log(4)}{\log(\text{IC50}/\text{IC80})}. \tag{7}$$

In cases where both IC50 and IC80 are identical ($h \to \infty$). In those cases, we set the Hill coefficient to a randomly selected hill coefficient taken from the distribution of all other isolates. Further, for these cases, the IC50 value is 'saturated' by experimental constraints. It may be that the virus is much more resistant, or that the real value of IC50 is much greater. Thus, in S2A Fig, we illustrate what these distributions might look like when saturated IC50s are drawn from a uniform distribution ranging from the saturated IC50 value up to $10^3$ $\mu$g/mL. Further, in Fig 3, we adjust the dose response curves to incorporate these theoretically possible resistant strains.

In S2B Fig we illustrate all PD data across virus clades.

## Modeling VRC01 pharmacodynamics

To incorporate VRC01 neutralization into our model, we employ a stoichiometric neutralization model [30]. To derive this model, we assume VRC01 creates immune complexes $C$, proportional to the concentration of free virus $V_f$ and the concentration of VRC01 $\mathcal{Y}$, given the rate constant $r$ as

$$C = r\mathcal{Y}^h V_f \tag{8}$$

where $h$ represents a stoichiometric factor indicating how many VRC01 molecules must be present to neutralize a single virion. The total amount of virus $V$ equates to the sum of the number of immune complexes and the number of free virions, that is $V = C + V_f$. Writing the fraction of bound virus as $v$ we have the relation

$$vV = r(1 - v)V\mathcal{Y}^h \tag{9}$$

which can be solved for the bound fraction as

$$v = \frac{r\mathcal{Y}^h}{1 + r\mathcal{Y}^h}. \tag{10}$$

We can then solve for a useful quantity, the concentration of VRC01 for which $v = 0.5$, or the 50% inhibitory concentration IC50 $= r^h$. Thus, the neutralized fraction (ranging from 0 to 1) at some time $t$ after infusion is

$$v(t) = \left[1 + \left(\frac{\text{IC50}}{\mathcal{Y}(t)}\right)^h\right]^{-1}. \tag{11}$$

Plots of this quantity after drawing random values for PK an PD are shown in Fig 3D. Then, during the stochastic viral dynamics simulation we interpret the reduction in free virus due to neutralization

$$V_f = [1 - v(t)]V \tag{12}$$

to model the protective impact of VRC01.

## Derivation of early infection basic reproductive number

The basic reproductive number $\mathcal{R}_0$ defines the average number of infected cells generated by an infected cell in a completely susceptible population. We derive an approximate basic reproductive number in the context of bNAb therapy from Eq 5 by employing several approximations. First, we assume the viral dynamics are much faster than cellular dynamics so that $\dot{V} \approx 0$ at all times and thus

$$V \approx \frac{\pi I_{A,P}}{\gamma}. \tag{13}$$

Similarly, the binding of antibodies is assumed to be rapid compared to infection dynamics such that the free virus is

$$V_f \approx [1 - v(t)]\frac{\pi I_{A,P}}{\gamma}. \tag{14}$$

We insert this expression into the differential equation for the active productively infected cells so that

$$\dot{I}_{A,P} = (1 - \lambda)\tau\beta S[1 - v(t)]\frac{\pi I_{A,P}}{\gamma} - \delta_I I_{A,P} - \kappa I_{A,P}E. \tag{15}$$

Next, we assume that early in infection the depletion of target cells is minimal, and the recruitment of adaptive cells has not occurred significantly. That is, we assume these cells remain at equilibrium $S = S^* = \alpha_S/\delta_S$ and $E = E^* = \alpha_E/\delta_E$. Inserting these fractions, the concentration of infected cells can be factored out to leave

$$\dot{I}_{A,P} = I_{A,P}\left\{(1 - \lambda)\tau\beta\frac{\alpha_S}{\delta_S}[1 - v(t)]\frac{\pi}{\gamma} - \delta_I - \kappa\frac{\alpha_E}{\delta_E}\right\}. \tag{16}$$

Dividing through by $\delta_I + \kappa\alpha_E/\delta_E$ allows us to identify the value

$$\mathcal{R}_0 = (1 - \lambda)\tau\beta\frac{\alpha_S}{\delta_S}\frac{\pi}{\gamma}\frac{1}{\left(\delta_I + \kappa\frac{\alpha_E}{\delta_E}\right)}. \tag{17}$$

As in the 'survival function' approach, we can think of this quantity as the product of the average lifespan of an active, productively infectious cell $[\delta_I + \kappa\alpha_E/\delta_E]^{-1}$, the rate at which virus is produced from this cell $\pi$, the average lifespan of these virions $1/\gamma$, d) the rate at which each virion infects susceptible cells to produce new infections $\beta\alpha_S/\delta_S$, and the fraction of these infections that are active and productive $(1 - \lambda)\tau$ [31].

We now define the time varying reproductive number that depends on VRC01 concentration and exposing strain as

$$\mathcal{R}_0(t) = \mathcal{R}_0[1 - v(t)], \tag{18}$$

such that the exponential solution to Eq 16 has the form

$$I_{A,P} \propto \exp\left[\frac{1}{T}(\mathcal{R}_0(t) - 1)\right]. \tag{19}$$

Therefore, if an exposure occurs when $\mathcal{R}_0(t) < 1$, the number of active, productively infectious cells decreases exponentially. Likewise, if $\mathcal{R}_0(t) > 1$ the number of infected cells grows

exponentially. This important threshold can be used to estimate the circumstances when exposure will result in infection.

## Modeling the infection probability during VRC01 infusion windows using the approximate early infection basic reproductive number

The basic reproductive number defines the stochastic extinction probability and therefore can be used to estimate the probability of infection, $1 - 1/\mathcal{R}_0$ [58]. However, in our model $\mathcal{R}_0(t)$ grows monotonically as VRC01 declines. Therefore, while in practice $\mathcal{R}_0(t) \leq 1$ at exposure typically results in extinction, extinction is not guaranteed.

Using the estimated parameters, we can calculate a range of possible values for $\mathcal{R}_0$ (see Fig 2C & 2D). Putting in values for $v(t)$ based on PK/PD estimates, the value of $\mathcal{R}_0(t)$ is plotted in Fig 3E. Then, to calculate the probability of infection, we calculated the fraction of simulated trajectories at every time for which $\mathcal{R}_0(t) > 1$. This is plotted in Fig 3F.

## Stochastic simulations of clinical trials

Using the ordinary differential equation system Eq 5, we developed a stochastic branching process simulation inspired by previous modeling of HIV [58, 59, 84]. Our implementation in Python, which employs the $\tau$-leap approach [85], is publicly available. In each time interval $\Delta t = 0.01$ days, a Poisson number of each transition type occurs. For instance, the number of susceptible cells grows $S = S + \Delta S$ where $\Delta S$ is a Poisson random variable $\mathcal{P}(\alpha_S \Delta t)$. Or, a certain number of active productively infected cells are created, $\Delta I_{A,P} = \mathcal{P}((1 - \lambda)\tau\beta S V_f \Delta t)$, which meanwhile removes the same number of susceptible cells and a free virus. When an active productively infectious cell dies, a poisson number of virions with mean $\pi$ are created. Such stochastic bursting has been shown to be similar to a model with continuous viral production [84].

The stochastic implementation allows us to simulate infection beginning with a single infected cell. Unlike the deterministic model, this approach permits burnout. Each of the initial state variable concentrations and rates involving concentration are converted to discrete values by multiplying by a volume. We choose this volume to be $10^8$ $\mu$L based on the observation that there is approximately 1-10 L of blood in an adult human and that there are approximately 10-100 times more T cells in lymph tissue than blood [86]. Our model for transmission is based on several elegant non-human primate studies [13, 15, 61, 87].

We assume that most sexual exposures to HIV are non-functional. In those exposures, sexual contact never results in infected cells in the recipient. This probabilistic process is not modeled, though it likely is responsible for the extremely rare per-coital transmission rate in humans [88]. Instead, we model functional exposures, defined as those in which a founder virus has crossed the mucosa and infected at least one local target cell in the recipient. These cells then are trafficked rapidly to lymph tissue where viral dynamics occurs.

To model the AMP trials, we assume repeated infusions of VRC01 every 8 weeks, and HIV viral load measurements (with detection threshold 30 copies/mL) every 4 weeks. Based on incidence data (and ignoring the influence of daily oral PrEP), we estimate approximately 40 functional exposures per trial arm [17]. This model does not distinguish between an individual who is doubly exposed and two individuals singly exposed. Then, for each 'trial' we simulated the three dosing arms, and computed prevention efficacy (PE) compared with placebo. We use a simplified calculation of PE using the ratio of infections that occur in a VRC01 treatment

arm $n^T$ to the number of infections in the placebo arm $n^P$. That is,

$$PE = 1 - \frac{n^T}{n^P}. \tag{20}$$

A more sophisticated estimator using a Nelson-Aalen estimator is described in [17]. Another formulation like the 'averted infections ratio' could be used alternatively [89]. Our simple measure can be interpreted as the total fraction of participants at the end of the trial that would likely have been infected but were instead protected due to VRC01. For example, if 7 individuals were infected in the 10 mg/kg dosing arm and 39 were infected in the placebo arm the prevention efficacy would be $1 - 7/39$, quoted as $PE = 82\%$.

## Global sensitivity analysis

Using the stochastic simulator of the clinical trials, we performed global sensitivity analyses on several key biological parameters (see Fig 5 and Table 1). For each simulated trial, we randomly chose values of all parameters simultaneously from ranges: potency reduction factor $\phi \in [1, 10^3]$, resistant fraction $f \in [0, 1]$, maximum IC50 $x_{50} \in [50, 10^4]$, initial number of infected cells $A_0 \in [0, 20]$. Variables covering several orders of magnitude were drawn uniformly on a logarithmic scale. Clinical outcomes including prevention efficacy, mean and variance of first positive viral loads, and mean and variance of breakthrough IC50s were recorded.

The resistant fraction $f$ parameterizes a simulated distribution of IC50s. The distribution is bi-modal with a fraction $f$ of IC50s drawn from a lognormal distribution centered on $IC50 = 10^3 \, \mu g/mL$ with variance 1 log and the remaining $1 - f$ drawn from a lognormal distribution centered on $IC50 = 10^{-2} \, \mu g/mL$ with 1 log variance. We vary $f$ from 0 to 1 because of we are uncertain how minor resistant variants may manifest in a realistic exposure.

## Clinical trial regression model (CTRM)

Based on the correlation between biological variables and measurable trial outcomes (Fig 5B), we developed a multi-variate regression model that we term a clinical trial regression model (CTRM) to estimate the sensitivity analysis variables ($v_j$) from the five clinical outcomes ($c_i$). That is, for each biological variable, we identified the regression coefficients $w_{ij}$ from the expression

$$v_j = \sum_i w_{ij} c_i + z_j + e_j. \tag{21}$$

where each $z_j$ is a constant and $e_j \sim \mathcal{N}(0, \sigma_j)$ is a normally distributed error term with a certain variance $\sigma_j^2$. The regression was accomplished using `sklearn` in Python.

The CTRM approach indicates that the full stochastic trial simulations can be reasonably well-approximated by a set of 5 regression coefficients for each sensitivity analysis variable.

## Supporting information

**S1 Fig. VRC01 pharmacokinetic (PK) data and model fitting.** A) VRC01 pharmacokinetic (PK) data: n = 12 individuals for each dose (color) with 3 infusions for each individual to demonstrate no meaningful accumulation occurs. B) The best-fit models to each individual participant infusion using Eq 6 in the main text. C) Estimated PK model parameters. Decay rates $k_1$, $k_2$ do not depend on infusion dose as expected.
(EPS)

**S2 Fig. Collected VRC01 pharmacodynamic (PD) data.** A) IC50s and IC80s collected from the LANL CATNAP database for VRC01 neutralization across all HIV clades. This allowed the Hill slope to be computed with Eq 7 in the main text; the median value is $h \sim 1$. To deal with saturated data where IC50 = IC80, we randomly assigned Hill slopes from non-saturated cases and also randomly drew IC50 values from the saturated value to a theoretical maximum of $10^3$ $\mu$g/mL. Thus, these distributions have substantial outliers. B) All raw pharmacodynamic neutralization characteristics separated by clade.
(EPS)

**S3 Fig. Raw breakthrough IC50 data from 100 simulated trials with varying potency reduction factor (A) and resistant fraction (B).** Potency reduction allows lower IC50 strains to infect, thus broadening the possible IC50s that can breakthrough. Increasing resistant fraction increases the number of breakthroughs with high IC50s.
(EPS)

## Acknowledgments

The authors would like to thank the many groups and individuals dedicated to planning and conducting the AMP trials (HVTN 704/08 + HVTN 703/HPTN 081). Moreover, all authors give their sincere gratitude to the trial participants. DBR thanks P Edlefsen, O Hyrien, A DeCamp, N Frahm, B Borate, L Zhang, D Montefiori, S Karuna, and other HVTN members for many valuable conversations. The views expressed are those of the authors and should not be construed to represent the positions of the U.S. Army, the Department of Defense, or the Department of Health and Human Services.

## Author Contributions

**Conceptualization:** Daniel B. Reeves, Elizabeth R. Duke, Joshua T. Schiffer.

**Data curation:** Daniel B. Reeves, Morgane Rolland, Merlin L. Robb.

**Formal analysis:** Daniel B. Reeves.

**Funding acquisition:** John R. Mascola, Myron S. Cohen, Lawrence Corey, Peter B. Gilbert.

**Methodology:** Daniel B. Reeves, Florencia A. Boshier, David A. Swan.

**Software:** Daniel B. Reeves.

**Supervision:** Yunda Huang, Bryan T. Mayer, E. Fabian Cardozo-Ojeda, Morgane Rolland, Peter B. Gilbert, Joshua T. Schiffer.

**Visualization:** Daniel B. Reeves.

**Writing – original draft:** Daniel B. Reeves.

**Writing – review & editing:** Daniel B. Reeves, Yunda Huang, Elizabeth R. Duke, Bryan T. Mayer, E. Fabian Cardozo-Ojeda, Myron S. Cohen, Peter B. Gilbert, Joshua T. Schiffer.

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
