## [Decision Letter · Decision Letter 0]

20 Oct 2019

Dear Dr Reeves,

Thank you very much for submitting your manuscript, 'Mathematical Modeling to Reveal Breakthrough Mechanisms in the HIV Antibody Mediated Prevention (AMP) Trials', to PLOS Computational Biology. As with all papers submitted to the journal, yours was fully evaluated by the PLOS Computational Biology editorial team, and in this case, by independent peer reviewers. The reviewers appreciated the attention to an important topic but identified some aspects of the manuscript that should be improved.

We would therefore like to ask you to modify the manuscript according to the review recommendations before we can consider your manuscript for acceptance. Your revisions should address the specific points made by each reviewer and we encourage you to respond to particular issues Please note while forming your response, if your article is accepted, you may have the opportunity to make the peer review history publicly available. The record will include editor decision letters (with reviews) and your responses to reviewer comments. If eligible, we will contact you to opt in or out.raised.

- Supporting Information uploaded as separate files, titled 'Dataset', 'Figure', 'Table', 'Text', 'Protocol', 'Audio', or 'Video'.

We hope to receive your revised manuscript within the next 30 days. If you anticipate any delay in its return, we ask that you let us know the expected resubmission date by email at ploscompbiol@plos.org.

Sincerely,

James Gallo

Associate Editor

PLOS Computational Biology

Rob De Boer

Deputy Editor

PLOS Computational Biology

[LINK]

Reviewer's Responses to Questions

**Comments to the Authors:**

Reviewer #1: In this study Reeves et al present a framework to predict and interpret the outcomes of trials for a new class of HIV therapeutics. These planned “Antibody mediated prevention” trials will test whether infrequent injections of synthetic anti-HIV antibodies can offer long-term prevention of new infections. This could be a paradigm shift in HIV prevention, since current prevention methods suffer from extreme difficulties in patient compliance (e.g. PrEP - taking antiretroviral pills daily as prophylaxis, or condom use, other behavioral modifications). There are two major open questions about these studies which Reeves et al address: What fraction of new infections are likely to be prevented by these antibodies and at what administered dose? And, if these trials are less successful than predicted, how can we understand what was the mechanism for failure?

Reeves et al develop and calibrate a modeling framework which includes a) pharmacokinetics of antibody in the blood, b) dose-dependent neutralizing potency of antibodies across a spectrum of different viral strains, c) the dynamics of acute HIV infection and its predicted dependence on % of free virus bound by antibody, and d) the frequency of HIV transmission and its predicted dependence on antibody administration at a population level. Overall the paper is a beautiful example of how modeling an integrate pre-clinical data from multiple sources to make informed predictions about new resource-intensive trials, and, how models can also be used to extract the most possible knowledge from the outcomes of these trials - be they successes or failures. The paper is very clearly written, the models and other methods well-justified, and the results presented in an intuitive way.

One main result of the paper is that the prevention efficacy of 10 mg/kg of VCR01 antibody administered every 8 weeks is 83%, while a 30 mg/kg dose is expected to have 92% efficacy. The second major result is that the two most likely modes of suboptimal prevention efficacy can be distinguished at a population scale if the antibody neutralization potency is measured (in vitro) for the virus of all individuals who get infected despite being on the therapy. The authors identify the two most likely mechanisms to comprise study efficacy: Firstly, the antibody may have a lower efficacy in vivo compared to in vitro, against all viral strains. This could be due to suboptimal pharmacokinetics in tissue sites relevant to infection or due to antibody competition. Secondly, there could be specific viral strains that are circulating which are very resistant to the antibody, but haven’t properly been catalogued in the database of strains the authors use for their main predictions. They show that these cases will both cause a reduction in study efficacy, but that the nature of viral strains that end up causing infections despite therapy will be different. If overall low potency is at play, then failure can be caused by strains with low IC50 (not very resistant), and there will be lots of variation in the IC50 of the strains that "break through". However if there are more circulating resistant strains and this causes poor prevention efficacy in the study, then when individuals do become infected, it is more likely to be caused by these very resistant strains, and there will be less variation in the IC50 of break through strains (uniformly high IC50).

Overall I think this paper would make an excellent addition to PLOS Computational Biology. The results will be directly and immediately useful to the four AMP trials that are beginning soon, and, to the many others that will likely follow. In addition, their modeling framework could also be applied to other modalities of prevention.

I only have a few suggestions and questions about the model. Most of these are relatively minor, though one I think is relatively serious since it is a clear mistake in the mathematics and will have some influence on the results. However, I think it is very easy to fix and does not change at all the overall framework the authors use.

Comments

1. R0 calculation: The formula for R0 that the authors give in the paper is wrong. This is caused by a fundamental misunderstanding of the underlying definition of R0. R0 generally provides the role of a threshold quantity in infection models, in that if R0<1 the infection will never take off or head towards extinction, whereas if R0>1 the infection will grow and reach a stable equilibrium where infection persists. However, there are infinitely many quantities that have this threshold behavior, but only one of them is equal to R0. There is only one specific expression that gives R0. R0 has the intuitive meaning as the average number of secondary infected cells produced by a single infected cell over the course of its lifespan, in a population that is otherwise at the uninfected equilibrium. In the author’s model, there are many types of cells, but since only the active/productive cells produce secondary infections, R0 should be the average number of secondary I_A,P cells produced from a single I_A,P cell. Many of these issues are discussed very clearly in the following paper: Heffernan JM, Smith RJ, Wahl LM, Perspectives on the basic reproductive ratio, JRSI, 2005. There are a few different ways to calculate R0 correctly and they will all give the same answer (different from the authors expression). I have attached documents showing both of these methods. The first (Derivation of R0.pdf), is often called the “survival function method” but is really just an accounting for the probabilities of survival and infection production throughout each stage of infection over the course of a generation. The second is the more formal next-generation matrix, and this calculation is done step by step in the attached Mathematica notebooks (Mathematica Output for R0.nb), from which I also printed off a pdf. Getting the expression for R0 right is very important for studies like this, because the impact that a given potency of therapy has on infection dynamics depends directly on how much it reduces R0 and whether that reduces R0 to less than one or not. If the wrong expression for R0 is used and R0 is assumed to be much smaller than it really is, predictions of treatment efficacy will be over optimistic. It is true that we can never be certain about the underlying model, but that is actually a bit tangential to the issue of estimating R0. At minimum R0 must be internally consistent within a model.

2. There is one second more minor issue related to R0. In general when estimating R0 from infection trajectories, R0 is determined from the early rate of exponential growth (r), and, importantly, the lifespan of infected cells (T). There is an intuitive explanation for this: the rate at which infection grows depends on how many offspring each infected cell is producing but also how long it takes to produce those offspring. In general there is some sort of expression like r ~ (R0-1)*/T. So what the authors say about there model being equally valid in it’s estimating of R0 to other models (e.g. Regoes 2010) despite different estimates of lifespan is simply not true. An accurate estimate of R0 requires an accurate estimate of lifespan. So R0 is only estimated correctly if the authors think their estimate of lifespan is correct.

3. I was a bit confused by the terminology of “productive” vs “unproductive” infection .. both of which can happen in either active or latently infected cells. Is this meant to refer to whether the viral genome is intact or defective? It’s hard to understand they way these cells behave in the model. How can active, unproductive cells stimulate the immune response (non-zero omega) but not produce free virus? And is it really necessary to track these cells?

4. The latent compartment has no influence on the kinetics observed during acute infection and therefore on the fit model - might be clearer just to say this, and that these parameters must be estimated from other types of data (on decay of LR in patients on ART and based on known fraction cells latently infected). This is different from other parameters, like Beta, alpha_s , pi, gamma, which were also fixed, but do strongly influence the acute infection kinetics, but are just mutually non-identifiable with other parameters.

5. With regards to the choice of parameters to be fit in the model: Some parts of this seemed quite weird to me. Is it really possible to identify four parameters all related to the immune response? That seems very unlikely to me. And how can it be possible to fit the early growth of infection in all patients with a fixed initial condition, especially since you can never know the actual time zero (time of infection). Considering fitting this upslope alone, like any linear fit, it should require two parameters - the intercept (initial condition, independent of model parameters used here), and slope (which depends on many model parameters but only two being fit, tau and delta_S). If you are forcing a particular intercept, this is likely skewing the estimate of the slope, and hence of R0. I also found it very strange that you would fix Beta based on another study and fit tau instead. Beta is the parameter that is the most difficult to directly measure in any way and therefore seems the oddest choice to fix. There are ways to measure tau and delta_I and \\delta_S and so they seem like much more natural choices. The study you cite for measuring Beta is from viral rebound after ART interruption, not acute infection, so it is very likely that Beta has a different value than in this study. And, these authors didn’t include unproductive infection, so their measured Beta would actually be Beta*tau in your model. Why not use another source to fix tau instead?

6. From the methods, it seems like the fitting was done on an individual level, so how are the population-level parameter distributions from Figure 2C generated?

7. For the many parameters that are fixed (Table 2), just adding a citation to a paper is not really a very good explanation for why you chose this value and a reader has no way of assessing the validity of this unless they go and read all these papers. It is important to many readers whether this parameter was actually measured mechanistically or just coming from another modeling group’s model fits. Would it be possible to include a paragraph in the methods or SI explaining all these sources?

8. In the main text you jump write from talking about a deterministic model to referring to a stochastic one without even a sentence or two describing the relationship between the two.

9. Figure 4: It doesn’t really make sense to report that a placebo has efficacy. If 40 functional exposures per year are observed (3% annual incidence) than you must make sure this comes out of your model. If some % of patients experience natural clearance, then the observed 40 exposures comes from True exposure rate * (1-% cleared). So, perhaps there are actually 45 exposures per year

10. Figure 5A: Why is there so much noise in these points? How big is each trial that was simulated? How long was the trial? This info should be in the caption.

11. Figure 6D: I think the c4 figure is wrong. Doesn’t match up with Figure 5A. As potency reduction factor goes up, strains with lower breakthrough IC50 should be able to get through. And as fraction resistant goes down, same thing. But this is opposite of Figure 6. Same with c5, standard deviation in IC50

12. SI: Some equation references are not working. Figure S2 panel labels wrong (don’t agree with caption)

Reviewer #2: The authors have constructed an elegant mathematical model to describe antibody-mediated HIV prevention. The model is particularly impressive because it incorporated a large amount of real world data from studies in which broadly neutralizing antibodies to the HIV envelope protein have been infused into study participants in a effort to prevent or control HIV infection. A key question addressed in the study is whether the limited efficacy seen in the prevention studies done to date reflects a lack of breadth (ability of the antibody to neutralize many different HIV variants) or a lack of potency (ability of the antibody to neutralize at different concentrations. The authors believe that from clinical trial data they can distinguish the cause of failure. This is important because it could guide future efforts to improve efficacy. The lack of breadth could be addressed by the choice of different antibodies while the lack of potency may require efforts to improve antibody pharmacokinetics.

Overall, this is a well written and potentially important study. I have only one concern. The authors emphasize the distinction between breadth and potency but in fact they are closely related. Antibodies that lack breadth simply have low potency for neutralizing diverse strains. I am not sure how this impacts the model, but some consideration of this concept should be included at least in the discussion.

Reviewer #3: The authors use a within-host infection model-based simulation strategy to develop a strategy to analyze clinical trial data from the AMP studies to gain insight into some mechanistic aspects of antibody-neutralization mediated protection. I especially liked the author's approaches of using clinical trial data for estimating parameter variability in their within-host infection models that were used for their downstream simulations, as well as using simulated clinical trial data to estimate unmeasurable mechanistic variables as a regression of measurable clinical trial outcomes. However, I do find the following major and minor concerns.

MAJOR CONCERNS:

1. Fig. 3 and analyses of pharmacodynamics: In the first part of the manuscript, the authors use data from the CATNAP database for VRC01 neutralization. It is not clear how viruses with both IC50 and IC80 greater than threshold are treated (the authors mention that Hill slope is median in such cases, but how is IC50 treated?). Such cases can arise due to two different scenarios: i) virus is neutralized (e.g. to 30% at highest concentration tested), but does not reach 50%, or ii) the virus lacks certain key mutations that make it completely resistant to the antibody (i.e. 0% neutralization). The latter case corresponds roughly to the "resistant" fraction used in the latter part of the paper. Looking at Fig 3B it seems that there are no viruses that fall in the latter category -- this does not seem likely to me. I urge the authors to address this issue and state their conventions clearly in the manuscript. Looking at only IC50 and IC80 will not be sufficient to distinguish the two scenarios, and in such cases, I recommend the use of fully resistant virus (scenario ii).

2. Lines 217-225: While the authors mention it in the Methods, I recommend describing in this section how the new distributions of viral sensitivity were chosen. It is not clear why the authors chose the particular values for the bimodal distribution, please explain. Also, given the second scenario in the above comment, the "resistant" IC50 centered around 10^3ug/ml would still give substantial neutralization in the early infusion cycle. Thus, I recommend the use of a higher threshold to characterize completely neutralization resistant viruses.

3. Clinical trial regression model: It is not clear if the authors have done a variable selection. It seems that the first time point viral load variables might not be adding much to the model, and it would be interesting to see how the predictions look with just the 3 other variables (protection efficacy and IC50 mean and std. dev.). I recommend the authors to explore this. Also, please use a more standard 5-fold cross validation instead of just one instance of a random 80-20 training-test split.

MINOR CONCERNS:

1. Lines 39-40: Several studies have shown that use of bnab combinations results in both increase of breadth and potency of neutralization. Thus, while the authors' statement is not incorrect, I find that the distinction between the scenarios is not a particularly important motivator. To me, a more important motivator is how can we estimate in vivo efficacy of antibody neutralization, for which an estimate of reduction of potency is paramount.

2. Lines 138-141: The naive interpretation that h~1 implies a single antibody binding a single virion is likely incorrect, given that multiple trimers exist on virions and most of them need to be inactivated by antibody binding for neutralization (see e.g. Brandenberg et al Trends in Microbiol 23(12) 763-774 (2015)). While higher stoichiometry binding lead to higher slopes, heterogeneity of rate constants across sub-populations in the genetically identical viral sample (coming from glycan heterogeneity, structural dynamics, etc.) can lead to an averaging of Hill curves that can reduce the slopes (similar to Fig. 3B) and lead to h~1. Thus, I urge the authors to reevaluate this statement, and discuss it appropriately.

3. Fig. 3D-E: I find the lines to be too thick and/or less transparent, and find that the average curves do not visually conform to the simulations (particularly the data shown in 3E vs 3F). I recommend using thinner or more transparent lines.

4. Lines 327-329: The authors can also mention that the regime in which all virions in an exposure are completely neutralized (i.e. 0 infected cells) will almost always result in R0 < 1, and is also, at least partially, captured by your model.

**Have all data underlying the figures and results presented in the manuscript been provided?**

Reviewer #1: No: Some data from a human clinical trial is not allowed to be shared publicly, special permissions required

Reviewer #2: Yes

Reviewer #3: Yes

PLOS authors have the option to publish the peer review history of their article (what does this mean?). If published, this will include your full peer review and any attached files.

Reviewer #1: No

Reviewer #2: No

Reviewer #3: No

---

## [Decision Letter · Decision Letter 1]

22 Dec 2019

Dear Dr Reeves,

We are pleased to inform you that your manuscript 'Mathematical Modeling to Reveal Breakthrough Mechanisms in the HIV Antibody Mediated Prevention (AMP) Trials' has been provisionally accepted for publication in PLOS Computational Biology.

In the meantime, please log into Editorial Manager at https://www.editorialmanager.com/pcompbiol/, click the "Update My Information" link at the top of the page, and update your user information to ensure an efficient production and billing process.

One of the goals of PLOS is to make science accessible to educators and the public. PLOS staff issue occasional press releases and make early versions of PLOS Computational Biology articles available to science writers and journalists. PLOS staff also collaborate with Communication and Public Information Offices and would be happy to work with the relevant people at your institution or funding agency. If your institution or funding agency is interested in promoting your findings, please ask them to coordinate their releases with PLOS (contact ploscompbiol@plos.org).

Thank you again for supporting Open Access publishing. We look forward to publishing your paper in PLOS Computational Biology.

Sincerely,

James Gallo

Associate Editor

PLOS Computational Biology

Rob De Boer

Deputy Editor

PLOS Computational Biology

Reviewer's Responses to Questions

**Comments to the Authors:**

Reviewer #1: The authors have carefully addressed all my comments, as well as those of the other reviewers.

Reviewer #3: No further comments.

**Have all data underlying the figures and results presented in the manuscript been provided?**

Reviewer #1: Yes

Reviewer #3: Yes

PLOS authors have the option to publish the peer review history of their article (what does this mean?). If published, this will include your full peer review and any attached files.

Reviewer #1: No

Reviewer #3: No

---

## [Editor Report · Acceptance letter]

13 Feb 2020

PCOMPBIOL-D-19-01414R1 

Mathematical Modeling to Reveal Breakthrough Mechanisms in the HIV Antibody Mediated Prevention (AMP) Trials

Dear Dr Reeves,

I am pleased to inform you that your manuscript has been formally accepted for publication in PLOS Computational Biology. Your manuscript is now with our production department and you will be notified of the publication date in due course.

With kind regards,

Laura Mallard
